# Exploring the Genetic Variability of *Gmelina arborea* Roxb. in Mexico with Molecular Markers to Establish an Efficient Improvement Program

**DOI:** 10.3390/plants14121888

**Published:** 2025-06-19

**Authors:** Marynor E. Ortega-Ramírez, Anuar Magaña-Álvarez, Daisy Pérez-Brito, Alberto Cortés-Velázquez, Ángel Nexticapan-Garcéz, Raúl Tapia-Tussell, Rodolfo Martín-Mex

**Affiliations:** 1Facultad de Ciencias Agropecuarias—CV, Universidad Autónoma de Chiapas, Carretera Ocozocoautla, Villaflores 30470, Chiapas, Mexico; marynor.ortega@unach.mx; 2GeMBio, Grupo de Estudios Moleculares Aplicados a la Biología, Centro de Investigación Científica de Yucatán A.C., Calle 43 # 130, x 32 y 34, Chuburná de Hidalgo, Mérida 97205, Yucatán, Mexico; anuar.magana@cicy.mx (A.M.-Á.); betocv@cicy.mx (A.C.-V.); angar@cicy.mx (Á.N.-G.); rodolfo@cicy.mx (R.M.-M.); 3Unidad de Energía Renovable, Centro de Investigación Científica de Yucatán A.C., Calle 43 # 130, x 32 y 34, Chuburná de Hidalgo, Mérida 97205, Yucatán, Mexico; rtapia@cicy.mx

**Keywords:** Melina (*Gmelina arborea*), molecular markers, genetic variability, simple primer amplification reaction (SPAR) markers

## Abstract

Melina (*Gmelina arborea* Roxb.) is a tree native to Asia, whose timber is not utilized in that region for a variety of reasons. However, the tree’s fast growth and extensive range of applications have increased its acceptance in other world’regions. *G. arborea* was introduced to Mexico in 1971, and it is currently the fifth most utilized forest species in commercial forest plantations (CFPs). However, its genetic diversity has not been evaluated in Mexico. The objective of this research was to investigate the genetic variability of Melina in Mexico using molecular markers. This investigation was undertaken to acquire valuable insights for the implementation of effective improvement strategies. A total of 85 Melina samples were collected from various locations in southeastern Mexico between 2017 and 2022. Genetic fingerprints were obtained using ten simple primer amplification reactions (SPARs): five Directed Amplification of Minisatellite DNA regions (DAMD), and five Inter-Simple Sequence Repeats (ISSRs). The polymorphic information content (PIC) was 0.940 and 0.950 for the DAMD and ISSR, respectively, and the similarity coefficients ranged from 0.12 to 0.88, indicating a high degree of polymorphism in the species under investigation. This is the first attempt to ascertain the genetic variability of *Gmelina arborea* in Mexico.

## 1. Introduction

The planting of forests is of great importance in the production of roundwood, fiber, bioenergy, and non-wood forest products. Additionally, forests provide a range of social and environmental services. In 2009, the Food and Agriculture Organization of the United Nations (FAO) published the “Global Planted Forests Outlook 2005–2030”, which reported that, despite comprising merely 7% of the global forest area, commercial forest plantations (CFPs) are responsible for at least 46% of global wood consumption [1].

By the end of 2023, Mexico had 138.7 million hectares of forest, with 48% of that area comprising wooded forest. Forestry production, amounting to 9.3 million cubic meters of roundwood, represents a mere 0.24% of the national gross domestic product (GDP). However, the trade balance indicates a deficit of USD 6274 million, as Mexico produces only approximately one-third of the wood products it consumes. Furthermore, the issue is exacerbated by the prevalence of gross deforestation, which results in the loss of approximately 200,000 hectares annually, as well as the impact of forest fires and the emergence of pests and diseases, which collectively affect an additional 350,000 hectares [2,3].

Considering the aforementioned factors, the Mexican government has, over the past few decades, actively encouraged the creation and growth of CFPs as a means of bolstering forest production, enhancing the yield and competitiveness of forest-based materials, and addressing the national deficit in forest resources. Furthermore, the government has endeavored to cultivate sustainable development in rural regions through the provision of alternative opportunities [2,3].

The fifth most utilized forest species in Mexican CFPs is Melina (*Gmelina arborea* Roxb.). As of the end 2024, this species was cultivated in 15 Mexican states, with a total area of 30,653 hectares within CFPs. The states with the largest planted areas are situated in humid and subhumid tropical regions, including Veracruz, Tabasco, Campeche, and Tamaulipas, in that order [4,5].

Melina is a medium-sized, fast-growing deciduous tree belonging to the Verbenaceae family, native to Asia. The tree is variously known as “white teak”, “Yemane”, “Coomb teak”, “cashmeri tree”, and “candhar tree”, depending on the region of cultivation. In its natural habitat in Asia, the tree attains a height of approximately 35 m and a diameter of more than three meters. In its juvenile stage, the tree displays smooth bark that ranges in color from whitish gray to yellow gray. Upon reaching maturity, the bark undergoes a color change, becoming darker with white mottling. Furthermore, the tree frequently displays multiple stems with a broad, spreading crown [6,7,8,9].

It is relatively uncommon to encounter extensive plantations of this species in Asia, largely due to the prevalence of insect and disease damage [10]. Moreover, the low density and light color of its wood render it uncompetitive in the Asian market, where other, denser and more valuable wood species are in greater demand. For these reasons, Melina has gained considerable acceptance as an exotic plantation species, largely due to its rapid growth and wide range of applications [6].

Melina was first introduced into the Americas between 1970 and 1975, with Brazil being the primary location of introduction [11]. In Mexico, the species was first introduced in 1971 by the National Institute of Forestry, Agriculture, and Livestock Research (INIFAP) in the “El Tormento” experimental field in Escárcega, Campeche. Subsequently, the species proliferated in other regions of the country, particularly in tropical areas [12].

This species has a wide range of applications, and it has generated interest due to its rapid growth and relatively short return on investment. The wood of *G. arborea* can be utilized in the production of a wide range of products, including pulp, furniture parts, timber, industrial timber, raw materials for solid products, and fodder [7,9]. Additionally, Melina is employed in the production of decorative veneers and in structural and light construction applications, including carpentry and joinery [12]. Various parts of the plant, including the roots, fruit, leaves, flowers and bark, are also used for medicinal purposes [8].

Some experts in forestry have suggested that Melina can be a valuable addition to agroforestry systems, provided that intensive cultivation is undertaken, and improved clones are employed to obtain more uniform wood, thereby enhancing the production of higher-quality products for the market and conferring greater resistance to pests and diseases. One of the key actions for this purpose is the development of new tree varieties that exhibit both desirable growth rates and dense wood. Hence, access to a diverse genetic repository of Melina is pivotal to address these challenges [6].

Most studies examining genetic variation in forest species have focused on quantitative phenotypic traits, employing progeny tests and provenance assays established in a range of environmental contexts [13]. However, molecular markers offer a more precise methodology for the study of phenotypic traits, particularly those techniques based on PCR. Such techniques permit direct sampling of the genome and accurate estimation of genetic variability between genotypes. Furthermore, they facilitate an understanding of the distribution and extent of genetic variation within and between species. This is beneficial for the implementation of effective and sustainable breeding and conservation strategies [13,14,15].

Single primer amplification reaction (SPAR) methods, including directed amplification of minisatellite DNA regions (DAMD), random amplification of polymorphic DNA (RAPD), and inter-simple sequence repeat (ISSR), have been employed for several years to examine genetic variation in plants. This is particularly for obtaining genetic fingerprints, especially for species for which such profiles have yet to be established, such as Melina [16].

Despite the availability of several molecular tools, to the best of our knowledge, an evaluation of the genetic diversity of *Gmelina arborea* has yet to be conducted. This may be because this species has not been deemed to possess significant economic value in its natural habitats where it competes with other species that are more highly valued [17]. Nevertheless, given its potential as a viable alternative for timber and related product production, it is crucial to ascertain the levels of genetic diversity in Melina.

The objective of this research was to investigate the genetic variability of *Gmelina arborea* Roxb. in Mexico using molecular markers. We aimed to obtain valuable insights into the genetic diversity of this species’ germplasm and establish the genetic relationships between accessions. The results of this study will be made available to producers and researchers, enabling the implementation of effective improvement strategies.

## 2. Results

### 2.1. Gmelina Arborea Plants and DNA Samples

A total of 85 samples of *G. arborea* plants were collected from seven different localities in three states, two of which (Veracruz and Tabasco) have the largest planted areas in Mexico. Five of these samples were sourced from San José, Costa Rica, and were reproduced in Mexico from in vitro clones (see Table 1, Figure 1 and Appendix A). It is important to note that all of the samples were utilized for the purposes of this study.

The genomic DNA of each sample was isolated and analyzed for quantity and quality. Their concentrations were in the range of 134 to 1224 ng/µL, with a mean of 902 ng/µL. The purity value of the DNAs (A260/A280 ratio) ranged from 1.65 to 2.19.

Furthermore, a 315-base pair (bp) PCR product was obtained for all samples when they were amplified for the chloroplast 16S ribosomal region. This finding indicated that there were no inhibitors for PCR reactions in these DNA samples.

### 2.2. SPAR Markers Analysis

In Table 2, the allelic status and PIC obtained using the molecular markers used are presented.

Tests with the two SPAR-type markers demonstrated mean polymorphism values for the primers that can be categorized as high in both instances (Appendix A).

As illustrated in Table 2, the majority of the DAMD markers demonstrated high PIC values (exceeding 0.97), except for M13, for which the polymorphic information content can be regarded as moderate to low. A comparable phenomenon was observed in the case of the ISSR markers. Four of the primers used exhibited PIC values exceeding 0.95, with the exception of IS14, which exhibited a PIC that, although it can be regarded as elevated, was lower than the remaining ISSR markers.

A dendrogram of the 85 samples under study was generated (Figure 2 and Appendix A) with data from the absence–presence matrices of the 10 SPARs markers. The analysis of the dendrogram demonstrates a broad spectrum of similarity among the totality of the samples, ranging from 0.12 to 0.88, which signifies a considerable degree of polymorphism between them. The samples that exhibited the highest genetic affinity were M5 and M6, both of which were collected in the Cárdenas locality, situated in Tabasco.

The sample designated EC1A4 (Emiliano Zapata, Tab.) exhibited the greatest genetic dissimilarity among the 85 samples analyzed, as evidenced by its similarity coefficient of 0.12 with respect to all other samples.

The remaining 84 samples exhibited a high degree of variability, as evidenced by the range of the similarity coefficient, which varied from 0.13 to 0.88, as depicted in the dendrogram (Figure 2). The initial group of samples to be formed comprises the in vitro plants from Costa Rica, propagated in the locality of Nacajuca, Tabasco (clade B). This group of plants (Fyt1 to Fyt5) demonstrated the lowest variability among them, with a similarity coefficient range from 0.76 to 0.86, thus separating them from the samples of Mexican origin (clade A) with which they showed little genetic relationship, as demonstrated by their low similarity coefficient (0.13).

Among the samples of Mexican origin, the group comprising samples M1 to M21 collected in the locality of Cárdenas, Tabasco (clade C), was of particular interest, as it demonstrated the highest genetic conservation with a similarity coefficient range of 0.6 to 0.88. As previously mentioned, the samples with the highest genetic relatedness (M5 and M6) were in this group. This clade demonstrated a low degree of genetic relatedness with respect to the remaining Melina plants (clade D), as indicated by their low similarity value (0.16).

Conversely, clade D exhibited the highest degree of genetic variability. The samples collected in Emiliano Zapata (Tabasco) were divided into two groups. The first group comprised samples EC1A21 to EC1A34 (clade E), which were separated from the rest of the plants, with a similarity coefficient of 0.2.

The second group of samples (clade F) was divided into three subgroups: the first subgroup (clade F2-I) consisted of samples from Emiliano Zapata (Tabasco) EC1A1 to EC1A20, which share a low genetic relationship (similarity coefficient of 0.23) with the secondary subgroup (clade F2-II) formed by samples from Nacajuca, Tabasco (H1 to H9); Moloacán, Veracruz (H10 to H12) and Reforma, Chiapas (Br1 to Br49). It is evident that both subgroups are in clade F2.

The third subgroup (clade F1) comprised samples from diverse geographical areas exhibiting shared genetic characteristics, including those from Emiliano Zapata, Tabasco (EC2A8, EC2A 10, and EC2A 11), Balancán, Tabasco (EC2A1, EC2A3, and EC2A 5), Salto de Agua, Chiapas (AG1 to AG5), and Palenque, Chiapas (AG6 to AG8). This F1 group, despite being classified within the same F clade as the F2 group, exhibited a low degree of genetic similarity (similarity coefficient of 0.2).

Figure 3 shows the principal coordinate analysis (PCoA) of the 85 *G. arborea* samples based on their molecular marker profiles, which distinguished between two distinct groups, A and B, based on their genetic similarity. All accessions were scattered according to their similarity. The first three components explained 40.6% of the variation.

As illustrated in Figure 3A, group A consists of samples from Cárdenas, Tabasco, which demonstrated the highest degree of genetic similarity. In contrast, Group B comprises the remaining samples.

Figure 3B shows the Principal Component Analysis in three dimensions. The analysis demonstrates that Group B, which was obtained in two dimensions, is further subdivided into two groups. Group B-1 comprises samples of Costa Rican origin cultivated in the locality of Nacajuca, Tabasco, which exhibit significant genetic dissimilarity to the other two groups. In addition, it can be observed that the samples from Cárdenas, Tabasco (group A) and the samples from Costa Rica (group B-1) exhibit a reduced genetic distance between them in comparison with group B-2, which comprises the remaining samples. The genetic variability between these groups is more pronounced, as evidenced by the increased dispersion of the points in the three-dimensional diagram.

## 3. Discussion

One of the fundamental premises for successful breeding is the acquisition of comprehensive knowledge about the available germplasm. In the context of forestry, where the species concerned are trees with a long-life cycle, this is imperative.

For many decades, molecular biology techniques have proven their usefulness in revealing the genetic variability between individuals and populations, making their application vital and necessary in the context of breeding programs for any crop [18].

To the best of our knowledge, *G. arborea* has not been studied using molecular tools. In this pioneering study of its genetic diversity in Mexico, it was demonstrated that molecular markers, specifically SPARs [16], are effective tools for exploring this diversity. Indeed, a combination of different types of markers is recommended to achieve a more accurate measurement of genetic diversity both between and within populations [14].

In this study, the markers used, DAMD and ISSR, facilitated the identification of a higher number of polymorphisms in the samples investigated, compared to other molecular markers, such as another type of SPAR: RAPD, which has been used in populations of the forest plant genus *Eucalyptus* [19,20], but whose accuracy has been questioned due to issues related to the reproducibility of the system [14].

The findings of our studies underline the importance of both the efficacy of the markers used in such research efforts and their selection based on their characteristics. This will more accurately estimate the genetic divergence and variability between genotypes, which is useful for implementing effective breeding strategies [13].

The genetic diversity observed among the *G. arborea* genotypes under investigation was found to be significantly higher than anticipated for these tree species planted in the three southeastern Mexican states.

It is generally accepted that, in the case of tree species, genetic diversity within populations is greater than between populations of the same species [14]. However, the results of this study demonstrate that there is greater diversity between populations than within them, with similarity coefficient values of only 0.12 being reached between the group located in Emiliano Zapata, Tabasco, and the rest of the samples analyzed. Even the closest populations in terms of genetic diversity were not so close, with values between 0.60 and 0.88 of similarity.

This phenomenon could be attributed to the potential origins of the Melina specimens introduced into Mexico, which may be diverse; some genotypes have come from Costa Rica and others from Brazil (personal communication from producers). Furthermore, upon their arrival in Mexico, these specimens were reportedly randomly planted in various states within the southeast region, where there is a higher production of this species. This could be corroborated by the high genetic difference between the accessions coming from Costa Rica and those already established in Mexico, some of which were believed by producers to have come from Costa Rica.

The genetic variation observed may also be attributable to nucleotide diversity within the genes responsible for the adaptive traits [14]. The environments of the three states in which the different accessions of this species are cultivated, although all tropical, exhibit slight variations in terms of rainfall and temperature. Some localities are between 100 and 300 km apart, as Balancán in Tabasco, and localities in Chiapas and Veracruz. It is also important to note that Palenque in Chiapas, and Balancán and Emiliano Zapata in Tabasco, are in the basin of the abundant Usumacinta River, while Cárdenas and Nacajuca, in Tabasco, are located within the basin of the Grijalva River, which creates small microclimates in these areas. Concomitant with the agronomic management conducted by the various local producers, these environmental factors may have contributed to the observed genetic variability.

The knowledge generated in this work will undoubtedly contribute to more informed and sound decisions on the improvement and management of *Gmelina arborea*.

## 4. Materials and Methods

### 4.1. Plant Material

Melina samples were collected between October 2017 and August 2022 in various locations in the south-southeastern region of Mexico. These locations were selected based on a prior analysis of the genetic variation observed by producers for the plant material utilized. Young leaves were harvested from adult plants that were in the process of full production. It was imperative that the material be free of any pathogens to prevent the introduction of foreign DNA sequences that could potentially interfere with the results.

### 4.2. DNA Isolation

Genomic DNA was extracted from the collected samples according to the plant DNA extraction protocol developed by Tapia-Tussell et al. (2005) at GeMBio [21].

The genomic DNA concentration and purity were determined by measuring the absorbance at 260 and 280 nm in a spectrophotometer (Nanodrop 2000, Madrid, Spain), and the DNA was diluted to a concentration of 20 ng/µL.

The quality and integrity of the DNA were evaluated via electrophoresis in a 0.8% (*w*/*v*) agarose gel, stained with ethidium bromide, and visualized in a UV light transilluminator (Bio-Rad Gel Doc Ez Imager, Hercules, CA, USA).

Furthermore, to ascertain that the Melina DNA samples were devoid of inhibitors, polymerase chain reaction (PCR) reactions were conducted with the 16S (forward and reverse) primers specific to chloroplasts (endogenous gene), which amplified a region of the 16S ribosomal gene, yielding a product of 315 base pairs (bp) [22].

### 4.3. SPAR Marker Analysis

The genetic diversity of Melina accessions was determined using ten SPAR markers, comprising five ISSR markers [23] and five DAMD markers [24] (Table 3).

For the ISSR markers, amplifications were developed in a final volume of 25 µL, containing 10 ng of genomic DNA, 1X PCR buffer (Invitrogen, Carslbad, CA, USA), 2.5 mM MgCl2 (Invitrogen, Carslbad, CA, USA), 0.25 mM dNTPs mix (Invitrogen, Carslbad, CA, USA), 1 µM of each primer, and 1U Taq DNA polymerase (Invitrogen, Carslbad, CA, USA). Amplification reactions were performed on a C1000 Touch thermal cycler (Bio-Rad, Hercules, CA, USA). The amplification conditions included initial denaturation for 2 min at 94 °C, followed by 36 cycles of amplification, each at 94 °C for 40 s, 54 °C for 45 s and 72 °C for 1.5 min, with a final extension step for seven minutes at 72 °C [23].

For DAMD markers, PCR reactions were performed as follows: the final volume of each reaction was 25 µL, containing 60 ng of genomic DNA, 1X PCR buffer (Invitrogen, Carslbad, CA, USA), 2 mM MgCl2 (Invitrogen, Carslbad, CA, USA), 0.25 mM dNTPs mix (Invitrogen, Carslbad, CA, USA), 0.2 µm primer, and 1U *Taq* DNA polymerase (Invitrogen, Carslbad, CA, USA). Amplification reactions were performed on a C1000 Touch thermal cycler (Bio-Rad, Hercules, CA, USA). The amplification conditions included initial denaturation for 5 min at 92 °C, followed by 40 cycles of amplification, each at 94 °C for 1 min, 55 °C for 2 min and 72 °C for two minutes with a final extension step for seven minutes at 72 °C [24].

Fragments obtained from the ISSR and DAMD markers were separated using both electrophoresis systems: acrylamide and agarose. In the case of acrylamide electrophoresis, the amplicons were separated at a concentration of 6% on polyacrylamide gels in a 29:1 ratio (acrylamide:bisacrylamide) (MilliporeSigma, St. Louis, MO, USA). The gels were run at a constant voltage of 200 V for 2.5 h in 1X Tris-Glycine (TG) buffer (MilliporeSigma, St. Louis, MO, USA). The DNA band sizes were determined by comparison with the 1 Kb Plus size molecular marker (Invitrogen, Carlsbad, CA, USA). Gel staining was performed using the silver staining technique [25]. In addition, PCR products were separated by electrophoresis in 1.5% (*w*/*v*) agarose (Invitrogen, Carslbad, CA, USA) gels; run at a constant voltage of 100 V for 1 h in 1X Tris-borate (TBE) buffer and were visualized by ethidium bromide staining (10 mg/mL) in a UV-Transilluminator (Bio-Rad, Hercules, CA, USA) [26].

### 4.4. Data Analysis

The PCR fingerprinting data were scored as discrete variables, with a value of “1” indicating the presence of a fragment and a value of “0” indicating the absence of a fragment. The binary data obtained by scoring the ISSR and DAMD profiles, obtained using different primers, both individually and cumulatively, were employed to construct a similarity matrix using Jaccard’s coefficients [25] and the unweighted pair group method. A dendrogram was generated using the unweighted pair group method with the arithmetic mean (UPGMA) and Jaccard’s similarity coefficient, with the sequential agglomerative hierarchical and nested (SAHN) clustering module of NTSYSpc software, version 2.02e [27], to facilitate the process. Principal coordinate analysis (PCoA) was also conducted to differentiate the oil palm accessions. The data were subsequently utilized to calculate the polymorphism information content (PIC) values in accordance with the equation proposed by Anderson et al. (1992) [28].

## 5. Conclusions

A high degree of genetic variability was identified among *G. arborea* trees from three southeastern Mexican states that were the subject of this study. The germplasm of this species cultivated in Mexico exhibits significant divergence from its counterpart in Costa Rica. Within the Mexican germplasm, the group located in Emiliano Zapata, Tabasco, was the most genetically distant (only 0.12 similarity coefficient) with respect to the rest of the processed samples, while the populations from Cardenas, Tabasco, had the highest genetic similarity between them, although with a range that can be considered medium to high. The application of molecular markers, namely DAMD and ISSR, proved to be highly effective in the identification of polymorphisms among the samples examined. The information presented here constitutes the first study to employ molecular markers in this forest species. Further studies that integrate the productive aspects of economic interests with morphological characteristics and molecular diversity should be conducted in *G. arborea*.

## Figures and Tables

**Figure 1 plants-14-01888-f001:**
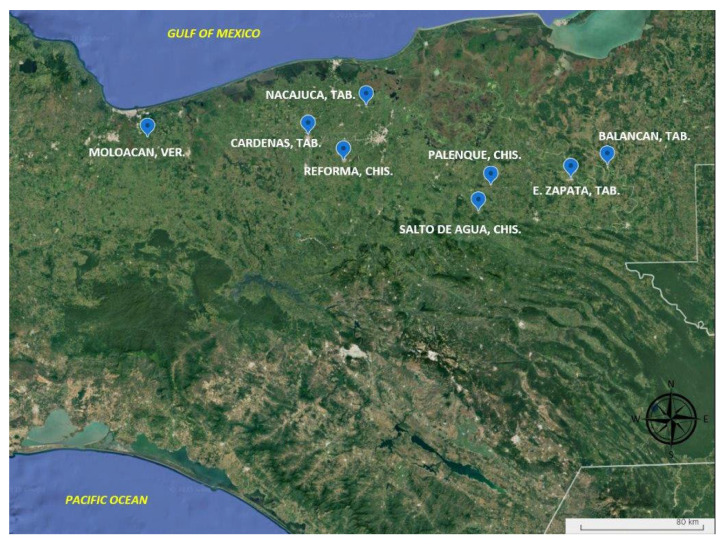
Geographical location of the collection sites of the *Gmelina arborea* in three states in the south and south-east of Mexico. VER: Veracruz state; TAB: Tabasco state; CHIS: Chiapas state.

**Figure 2 plants-14-01888-f002:**
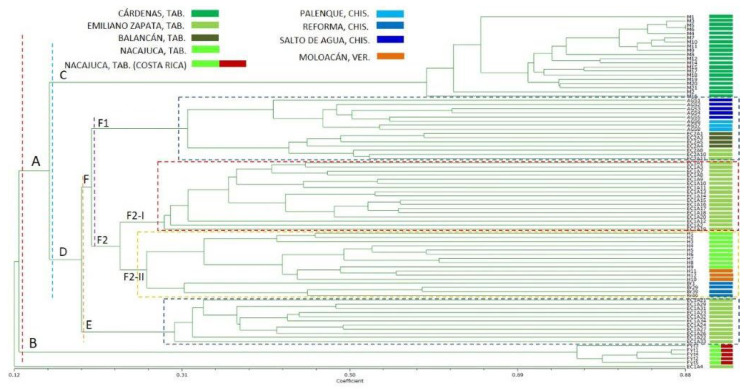
DAMD and ISSR consensus of the unweighted pair-group arithmetic average (UPGMA) dendrogram based on Jaccard’s coefficient, showing the relationships between 85 *G. arborea* plant accessions determined using cumulative data. VER: Veracruz state; TAB: Tabasco state; CHIS: Chiapas state.

**Figure 3 plants-14-01888-f003:**
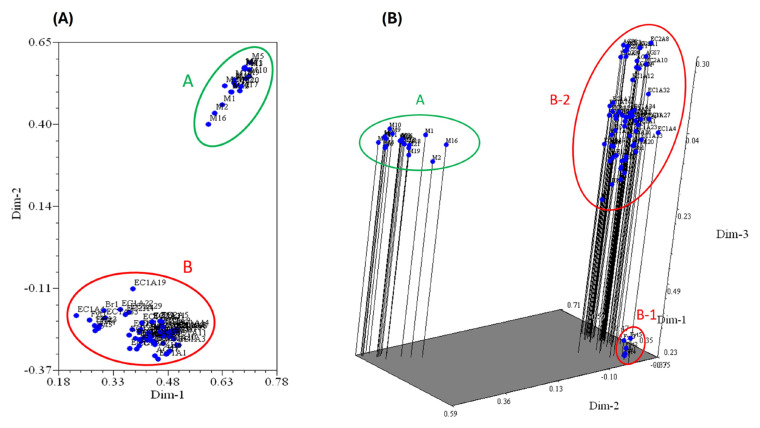
Principal coordinate analysis obtained via NTSYSpc software, version 2.02e (Rohlf 2000): (**A**) two-dimensional representation; (**B**) Three-dimensional representation. The letters A (in green) and B (in red) are the names of the sample groupings in each case.

**Table 1 plants-14-01888-t001:** Data from the collection of *G. arborea* samples in Mexico, including the state, municipality and the number of samples obtained.

Location Key	State	Municipality	Number of Samples
M1-M12, M14-M21	Tabasco	Cárdenas	20
AGS1-AGS5	Chiapas	Salto de Agua	5
AGS6-AGS8	Chiapas	Palenque	3
EC1A1-EC1A4; EC1A7-EC1A24; EC1A26-EC1A27; EC1A29; EC1A31-EC1A34	Tabasco	Emiliano Zapata	29
EC2A1; EC2A3-EC2A5; EC2A8; EC2A10-EC2A11	Tabasco	Balancán	7
H1–H9	Tabasco	Nacajuca	9
H10-H12	Veracruz	Moloacán	3
Br01; Br29; Br39; Br49	Chiapas	Reforma	4
Fyt1-Fyt5 *	Tabasco	Nacajuca	5
Total			85

* This material was obtained from in vitro germplasm sourced from San Jose, Costa Rica.

**Table 2 plants-14-01888-t002:** Allelic status and polymorphic information content (PIC) obtained via the molecular markers used in this work.

SPARMarkers	Primer Name	Number of Amplified Bands	Number of Polymorphic Bands	Allele Size	High Frequency Allele		PIC
				Range (bp)	Size (bp)	Frequency (%)	
DAMD	33.6	98	98	220–3700	1650	51	0.973
	HVV	70	70	330–4000	1350	47	0.978
	HVA	71	71	300–3200	575	60	0.973
	HVR	75	75	340–3900	975	43	0.986
	M13	29	29	770–4700	4700 3700 2800 1520 1190	64	0.789
Total	-	343	343	220–4700	-	-	-
Average	-	68	68	-	-	-	0.940
ISSR	IS01	83	83	160–3800	1300 320	51	0.958
	IS13	59	59	300–3000	950	67	0.974
	IS14	57	56	170–2000	690	100	0.886
	IS17	85	85	126–3850	1050	76	0.966
	IS19	100	100	180–2700	330	55	0.968
Total	-	384	383	126–3850	-	-	-
Average	-	76	76	-	-	-	0.950

**Table 3 plants-14-01888-t003:** SPAR markers that were used in this study [23,24].

Primer Name	Sequence 5′-3′
ISSR	
IS01	(GACA)_4_
IS13	(CT)_8_TG
IS14	(AG)_8_TA
IS17	(TG)_8_GT
IS19	(AG)_8_TC
DAMD	
HBV	GGTGTAGAGAGGGGT
M13	GAGGGTGGCGGTTCT
HVA	AGGATGGAAAGGAGGC
33.6	AGGGCTGGAGG
HVR	CCTCCTCCCTCCT

## Data Availability

The data within this paper is available from the corresponding author upon reasonable request.

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
