# Peer review of "Exploring the Genetic Variability of Gmelina arborea Roxb. in Mexico with Molecular Markers to Establish an Efficient Improvement Program"

_plants, 2025, doi:10.3390/plants14121888_

Round 1
Reviewer 1 Report
Comments and Suggestions for Authors
The paper conducts a comprehensive study on Melina, an economically important forest tree introduced from Asia, utilizing DAMD and ISSR molecular markers. Molecular marker experiments were performed on 85 samples collected from different regions, and classification was conducted based on the experimental data. This research holds significance for understanding the geographical distribution patterns and population diversity of Melina in Mexico. The following issues require clarification:
- The sampling proportions from the Cárdenas and Emiliano Zapata regions are notably large. Could this disproportionate sampling bias the overall estimates, and is there a possibility of close reproductive relationships among these samples?
- The analysis reveals that the vast majority of marker loci exhibit high PIC values, indicating extremely rich genetic diversity. Such high indices in an introduced plant species are remarkable—is this related to the scale of introduction or other factors?
- The paper mentions the stability of chloroplast PCR fragment lengths. Do the highly variable regions of the chloroplast also support the classification criteria established by the nuclear genome?
Author Response
Response to Reviewer 1
Comments and Suggestions for Authors
The paper conducts a comprehensive study on Melina, an economically important forest tree introduced from Asia, utilizing DAMD and ISSR molecular markers. Molecular marker experiments were performed on 85 samples collected from different regions, and classification was conducted based on the experimental data. This research holds significance for understanding the geographical distribution patterns and population diversity of Melina in Mexico. The following issues require clarification:
- The sampling proportions from the Cárdenas and Emiliano Zapata regions are notably large. Could this disproportionate sampling bias the overall estimates, and is there a possibility of close reproductive relationships among these samples?
R/ The discrepancy in the number of samples obtained from the municipalities of Cárdenas and Emiliano Zapata in comparison to other localities can be attributed to the presence of a significant number of Gmelina plantations in these specific areas. The number of samples obtained is directly proportional to the number of plantations present, thereby ensuring the absence of any bias. Furthermore, the possibility of close reproductive relationships between the samples is negated, as evidenced by the substantial geographical distance between the two groups, as illustrated in Figure 1.
- The analysis reveals that the vast majority of marker loci exhibit high PIC values, indicating extremely rich genetic diversity. Such high indices in an introduced plant species are remarkable—is this related to the scale of introduction or other factors?
R/ We consider that the variability found in the plantations may be due to two factors. Firstly, the plantations have different origins, as they are managed by various companies that have introduced germplasm from different countries, mainly Costa Rica and Brazil. This is similar to what has happened with other forest species in Mexico, such as eucalyptus and oil palm, which explains the significant variability observed. Secondly, it has been noted that the clones planted are greatly influenced by the environment of the different localities, as they are the result of many years of trials in each municipality. While all these localities have a tropical climate, there are differences in temperature and rainfall. Some localities also have microclimates due to their proximity to the Usumacinta and Grijalva river basins. These conditions may affect the expression of some genes.
- The paper mentions the stability of chloroplast PCR fragment lengths. Do the highly variable regions of the chloroplast also support the classification criteria established by the nuclear genome?
R/ Indeed, the highly variable regions of the chloroplast can support the classification criteria established by the nuclear genome, albeit not necessarily in the same way. The chloroplast genome, despite its independence from the nuclear genome, furnishes complementary information that is instrumental in taxonomic and phylogenetic studies. Furthermore, the primers utilized in this study developed by Trejo-Saavedra et al., 2015 [23] were designed for the standardization of PCR analysis to amplify the chloroplast 16S ribosomal ribonucleic acid gene. This gene is highly conserved in plants, and the reaction will indicate whether the concentration and condition of the DNA is appropriate before proceeding to detection.

Reviewer 2 Report
Comments and Suggestions for Authors
In the paper “ Exploring with molecular markers the genetic variability of 2 Gmelina arborea Roxb in Mexico to stablishing an efficient improvement program”, the authors evaluate 85 samples of Melina, which were collected from various locations in southeastern Mexico, between the years, 2017 and2022, using various molecular marker techniques They report a high degree of polymorphism, among the samples analyzed and refer to this study as the first report on genetic variability in tres of Melina in Mexico.
The paper in general is concise , well presented, with the appropriate references, however, I suggest considering some aspects before approving the article as followed
- Inthe introduction section, the authors refer to statistical, productive and economic datas related to the wood industry and include various indicators in terms of extension and economic return, however, it is not clear to this reviewer the period of time from which these data were obtained, since they vary from year to year, depending on various conditions, both market and environmental and productive.
- Inthe materials and methods section, it is important that the authors properly cite the DNA extraction protocol, as they refer to a protocol that was developed in a laboratory, however, the correct way is to include the corresponding citation of the author who developed the protocol.
- Inthat same section, I suggest to include a complete section of reproducibility analysis of each of the techniques used, since for visualization, they used silver staining, which is a technique that can show a variable number of bands, depending on the experimental conditions of the development, leading to the appearance of ambiguous bands that could bias the analysis between the samples.
- Next, I suggest making clear emphasis on the percentage of reproducibility of the various techniques.
- Inthe results section, I suggest changing the orientation of figure two to increase its size and definition, since there are groups in which it is difficult to visualize the samples corresponding to that group, although it is well described in the text, this modification would help the correct interpretation of the figure.
- Inthis section (Results), it is important to include a small discussion of possible geographical barriers, between the most contrasting regions that complement the wide explanation of the genetic diversity found, with emphasis on the most variable groups or individuals.
In general, the article is well written, although a general review of the grammatical structure of English would be useful.
These changes, from my point of view, would enrich the final version of the article, which for this reviewer is an article that should be accepted after making these proposed changes.
Comments on the Quality of English Languagegeneral review of the grammatical structure of English would be useful.
Author Response
Responses to Reviewer 2
Comments and Suggestions for Authors
In the paper “ Exploring with molecular markers the genetic variability of 2 Gmelina arborea Roxb in Mexico to stablishing an efficient improvement program”, the authors evaluate 85 samples of Melina, which were collected from various locations in southeastern Mexico, between the years, 2017 and2022, using various molecular marker techniques They report a high degree of polymorphism, among the samples analyzed and refer to this study as the first report on genetic variability in tres of Melina in Mexico.
The paper in general is concise, well presented, with the appropriate references, however, I suggest considering some aspects before approving the article as follows:
- In the introduction section, the authors refer to statistical, productive and economic datas related to the wood industry and include various indicators in terms of extension and economic return, however, it is not clear to this reviewer the period of time from which these data were obtained, since they vary from year to year, depending on various conditions, both market and environmental and productive.
R/ The dates and time periods referenced in the initial two paragraphs have been incorporated into the text in the introduction section (lines 38-40, 42 and 59).
- In the materials and methods section, it is important that the authors properly cite the DNA extraction protocol, as they refer to protocol that was developed in a laboratory, however, the correct way is to include the corresponding citation of the author who developed the protocol.
R/ The reference to the authors of the protocol was correctly incorporated into the Materials and Methods section, on line 335.
- In that same section, I suggest to include a complete section of reproducibility analysis of each of the techniques used, since for visualization, they used silver staining, which is a technique that can show a variable number of bands, depending on the experimental conditions of the development, leading to the appearance of ambiguous bands that could bias the analysis between the samples.
R/ As a research and service laboratory, we have utilized silver staining to visualize PCR products on polyacrylamide gels for an extended period, and the reproducibility of this technique has been consistently reliable. This approach has enabled us to establish consistent banding patterns for the genetic fingerprint of different species, with no observed variation over time and between patterns. The authors consider that the reproducibility of the techniques used in this study is well documented in several research articles in high-impact scientific journals, and that devoting a whole section within the manuscript to this topic would distract from the study's primary objective. However, we have listed some references on the usefulness and reproducibility of ISSR and DAMD, as well as the advantages of the silver staining technique for revealing PCR products in a polyacrylamide electrophoresis system. We present these to the reviewer so that he (or she) knows what informed our selection of techniques:
Alatar, A. A., Faisal, M., Abdel-Salam, E. M., Canto, T., Saquib, Q., Javed, S. B., ... & Al-Khedhairy, A. A. (2017). Efficient and reproducible in vitro regeneration of Solanum lycopersicum and assessment genetic uniformity using flow cytometry and SPAR methods. Saudi Journal of Biological Sciences, 24(6), 1430-1436.
Purayil, F. T., Robert, G. A., Gothandam, K. M., Kurup, S. S., Subramaniam, S., & Cheruth, A. J. (2018). Genetic variability in selected date palm (Phoenix dactylifera L.) cultivars of United Arab Emirates using ISSR and DAMD markers. 3 Biotech, 8, 1-8.
Vanishree, G., Patil, V. U., Kardile, H., Bhardwaj, V., Singh, R., & Chakrabarti, S. K. (2016). DNA fingerprinting of Indian potato cultivars by inter simple sequence repeats (ISSRS) markers. Potato Journal, 43(1).
Ng, W. L., & Tan, S. G. (2015). Inter-simple sequence repeat (ISSR) markers: are we doing it right. ASM Sci J, 9(1), 30-39.
Pakseresht, F., Talebi, R., & Karami, E. (2013). Comparative assessment of ISSR, DAMD and SCoT markers for evaluation of genetic diversity and conservation of landrace chickpea (Cicer arietinum L.) genotypes collected from north-west of Iran. Physiology and Molecular Biology of Plants, 19, 563-574.
Bassam, B. J., & Gresshoff, P. M. (2007). Silver staining DNA in polyacrylamide gels. Nature protocols, 2(11), 2649-2654.
- Next, I suggest making clear emphasis on the percentage of reproducibility of the various techniques.
R/ The techniques employed in this study are among the most widely utilized molecular markers on an international scale, a consequence of their exceptional reproducibility. It should be noted that, although the RAPD markers are within the group called SPARs, they were not utilized in the present study, precisely because it has been highlighted on an international level that there may be issues with their reproducibility, but not with the markers that were employed. In the discussion section (lines 283-288), reference is made to this.
- In the results section, I suggest changing the orientation of figure two to increase its size and definition, since there are groups in which it is difficult to visualize the samples corresponding to that group, although it is well described in the text, this modification would help the correct interpretation of the figure.
R/ It was not possible to alter the orientation of Figure 2, because the computer programme from which it was obtained provides it in that form. Furthermore, given the number of samples included in the dendogram, it is not considered that the names would be much more visible even if they could be changed.
- In this section (Results), it is important to include a small discussion of possible geographical barriers, between the most contrasting regions that complement the wide explanation of the genetic diversity found, with emphasis on the most variable groups or individuals.
R/The requested discussion of possible geographical barriers between the most contrasting regions, to complement the explanation of the genetic diversity found with emphasis on the most variable groups or individuals, was included in the Discussion section (lines 315 –320).

Reviewer 3 Report
Comments and Suggestions for Authors
The study provides a detailed analysis of the genetic variability of Gmelina arborea in Mexico using molecular markers, offering valuable insights into the genetic diversity of this species in the region.
The main question addressed in this research is to investigate the genetic variability of Gmelina arborea Roxb in Mexico using molecular markers of DAMD and ISSR.
The topic is both original and relevant to the field of forestry and genetic diversity research. It addresses a specific gap by being the first study to explore the genetic diversity of Gmelina arborea Roxb in Mexico. As mentioned in the paper, most studies examining genetic variation in forest species have focused on quantitative phenotypic traits. This study, however, utilizes molecular markers to explore genetic diversity, allowing for a accurate estimation of genetic variability between genotypes.
Compared with other published material, this study adds the first exploration of the genetic variability of Gmelina arborea Roxb in Mexico. It provides a baseline for understanding the genetic diversity of this species in the country and offers insights into the genetic relationships between accessions. The application of molecular markers contributes to the field by demonstrating their effectiveness in identifying polymorphisms in Gmelina arborea. This information is crucial for the development of effective and sustainable breeding and conservation strategies.
The conclusions are largely consistent with the evidence and arguments presented. The study demonstrates a high degree of genetic variability among Gmelina arborea plantations in three southeastern Mexican states. This finding is supported by the results of the SPAR marker analysis, which showed high polymorphism information content (PIC) values and a broad range of similarity coefficients. The conclusions also address the main question posed by the research by highlighting the significance of the genetic diversity findings for the implementation of effective improvement strategies.
The references are nearly appropriate. Apart from the fact that most of the literature is old, they include key papers on genetic diversity in forest species, the use of molecular markers in genetic studies, and other relevant topics.
The authors could consider the following improvements regarding the methodology:
1. Phenotypicdata: Integrating phenotypic data with the genetic data would provide a more comprehensive understanding of the genetic diversity and its implications for improvement programs. This could involve assessing traits such as growth rate, wood quality, and disease resistance.
2. Additional molecular markers: While SPAR markers were effective in this study, the use of additional molecular markers such as simple sequence repeats (SSRs) or single nucleotide polymorphisms (SNPs) could provide even more detailed information about the genetic diversity.
3. Sample Size: Increasing the number of samples to include more individuals from a wider range of locations would enhance the representativeness of the study and provide a more comprehensive picture of the genetic diversity of Gmelina arborea in Mexico. only 85 samples were collected, this may not be sufficient to fully capture the entire genetic diversity of Gmelina arborea across all of Mexico. More extensive sampling could provide a more comprehensive understanding.
In addition, the paper does not integrate molecular diversity data with other relevant information such as growth performance, wood quality, or resistance to diseases and pests. Such integration could enhance the practical application of the findings in breeding programs.
Additional comments on the tables and figures: some of the figures could be improved by adding more detailed labels and explanations to enhance their clarity.
Author Response
Response to Reviewer 3
Comments and Suggestions for Authors
The study provides a detailed analysis of the genetic variability of Gmelina arborea in Mexico using molecular markers, offering valuable insights into the genetic diversity of this species in the region.
The main question addressed in this research is to investigate the genetic variability of Gmelina arborea Roxb in Mexico using molecular markers of DAMD and ISSR.
The topic is both original and relevant to the field of forestry and genetic diversity research. It addresses a specific gap by being the first study to explore the genetic diversity of Gmelina arborea Roxb in Mexico. As mentioned in the paper, most studies examining genetic variation in forest species have focused on quantitative phenotypic traits. This study, however, utilizes molecular markers to explore genetic diversity, allowing for a accurate estimation of genetic variability between genotypes.
Compared with other published material, this study adds the first exploration of the genetic variability of Gmelina arborea Roxb in Mexico. It provides a baseline for understanding the genetic diversity of this species in the country and offers insights into the genetic relationships between accessions. The application of molecular markers contributes to the field by demonstrating their effectiveness in identifying polymorphisms in Gmelina arborea. This information is crucial for the development of effective and sustainable breeding and conservation strategies.
The conclusions are largely consistent with the evidence and arguments presented. The study demonstrates a high degree of genetic variability among Gmelina arborea plantations in three southeastern Mexican states. This finding is supported by the results of the SPAR marker analysis, which showed high polymorphism information content (PIC) values and a broad range of similarity coefficients. The conclusions also address the main question posed by the research by highlighting the significance of the genetic diversity findings for the implementation of effective improvement strategies.
The references are nearly appropriate. Apart from the fact that most of the literature is old, they include key papers on genetic diversity in forest species, the use of molecular markers in genetic studies, and other relevant topics.
The authors could consider the following improvements regarding the methodology:
- Phenotypic data: Integrating phenotypic data with the genetic data would provide a more comprehensive understanding of the genetic diversity and its implications for improvement programs. This could involve assessing traits such as growth rate, wood quality, and disease resistance.
R/ Integrating phenotypic and genotypic data would undoubtedly have made the study more complete. However, as the project funding this research focused on the molecular aspect, it was not possible to obtain phenotypic data from the sampled plantations. Therefore, they were not included in the study. However, since the main objective was to take a preliminary look at the molecular variation of Gmelina, we considered that a more in-depth study of phenotype and genotype could be conducted using the information obtained in this initial study.
- Additional molecular markers: While SPAR markers were effective in this study, the use of additional molecular markers such as simple sequence repeats (SSRs) or single nucleotide polymorphisms (SNPs) could provide even more detailed information about the genetic diversity.
R/ Undoubtedly, an increase in the number of markers of different types used in any study of molecular variability will result in a greater amount of genomic information being obtained. In this study, as previously outlined, a comprehensive review of the existing literature was conducted, revealing a paucity of research on the molecular information in Gmelina. Consequently, the decision was made to utilize the two types of markers, ISSR and DAMD, which have been widely recommended in numerous studies for this research objective, namely the analysis of molecular profiles that have not been previously obtained. It is hypothesized that subsequent breeding studies will be able to identify SSR and SNP markers, that have been utilized in other species from nearby families, if not from the same Gmelina family, for use in this species.
- Sample Size: Increasing the number of samples to include more individuals from a wider range of locations would enhance the representativeness of the study and provide a more comprehensive picture of the genetic diversity of Gmelina arborea in Mexico. only 85 samples were collected, this may not be sufficient to fully capture the entire genetic diversity of Gmelina arborea across all of Mexico. More extensive sampling could provide a more comprehensive understanding.
R/ The project that financed the study was only proposed for southern Mexico, with a particular focus on the states of Tabasco, Chiapas and Campeche. These states account for 60% of Mexico's planted areas of this tree and have the greatest potential for future planting of this forest species. This is why samples were collected from these specific states. The assumption was made that all individuals within a given plantation were highly similar, since the sampled plantations all came from clones. The discrepancy in the samples obtained from the various locations can be attributed to the variation in the size of the plantations. For the purposes of our study, we consider that this number of samples was sufficient, as it revealed that the genetic diversity among populations in areas of interest is much higher than expected and serves as a basis for more extensive studies in the near future.
In addition, the paper does not integrate molecular diversity data with other relevant information such as growth performance, wood quality, or resistance to diseases and pests. Such integration could enhance the practical application of the findings in breeding programs.
R/ As explained above, the project was funded for the molecular study, so it was not possible to include phenotypic variables.
Additional comments on the tables and figures: some of the figures could be improved by adding more detailed labels and explanations to enhance their clarity.
R/ The figures have been adjusted to make them clearer.

Round 2
Reviewer 3 Report
Comments and Suggestions for Authors
The author has basically responded to the questions I raised, and I have no further suggestions.